# An Unsupervised Deep Learning Approach for Real-World Image Denoising

**Dihan Zheng[1], Sia Huat Tan[2], Xiaowen Zhang[3], Zuoqiang Shi[4,5], Kaisheng Ma[2], Chenglong Bao[1,5]\***

[1]Yau Mathematical Sciences Center, Tsinghua University
[2]Institute for Interdisciplinary Information Science, Tsinghua University    [3]Hisilicon
[4]Department of Mathemathcal Sciences, Tsinghua University
[5]Yanqi Lake Beijing Institute of Mathematical Sciences and Applications
{zhengdh19,csf19}@mails.tsinghua.edu.cn, zhangxiaowen9@hisilicon.com,
{zqshi,kaisheng,clbao}@mail.tsinghua.edu.cn

## Abstract

Designing an unsupervised image denoising approach in practical applications is a challenging task due to the complicated data acquisition process. In the real-world case, the noise distribution is so complex that the simplified additive white Gaussian (AWGN) assumption rarely holds, which significantly deteriorates the Gaussian denoisers' performance. To address this problem, we apply a deep neural network that maps the noisy image into a latent space in which the AWGN assumption holds, and thus any existing Gaussian denoiser is applicable. More specifically, the proposed neural network consists of the encoder-decoder structure and approximates the likelihood term in the Bayesian framework. Together with a Gaussian denoiser, the neural network can be trained with the input image itself and does not require any pre-training in other datasets. Extensive experiments on real-world noisy image datasets have shown that the combination of neural networks and Gaussian denoisers improves the performance of the original Gaussian denoisers by a large margin. In particular, the neural network+BM3D method significantly outperforms other unsupervised denoising approaches and is competitive with supervised networks such as DnCNN, FFDNet, and CBDNet.

## 1 Introduction

Noise always exists during the process of image acquisition and its removing is important for image recovery and vision tasks, e.g., segmentation and recognition. Specifically, the noisy image $\mathbf{y}$ is modeled as $\mathbf{y} = \mathbf{x} + \mathbf{n}$, where $\mathbf{x}$ denotes the clean image, $\mathbf{n}$ denotes the corrupted noise and image denoising aims at recovering $\mathbf{x}$ from $\mathbf{y}$. Over the past two decades, this problem has been extensively explored and many works have been proposed.

Among these works, one typical kind of model assumes that the image is corrupted by additive white Gaussian noise (AWGN), i.e., $\mathbf{n} \sim \mathcal{N}(0, \sigma^2\mathbf{I})$ where $\mathcal{N}(0, 1)$ is the standard Gaussian distribution. Representative Gaussian denoising approaches include block matching and 3D filtering (BM3D) (Dabov et al., 2007b), non-local mean method (NLM) (Buades et al., 2005), K-SVD (Aharon et al., 2006) and weighted nuclear norm minimization (WNNM) (Gu et al., 2014), which perform well on AWGN noise removal. However, the AWGN assumption seldom holds in practical applications as the noise is accumulated during the whole imaging process. For example, in typical CCD or CMOS cameras, the noise depends on the underlying context (daytime or nighttime, static or dynamic, indoor or outdoor, etc.) and the camera settings (shutter speed, ISO, white balance, etc.). In Figure 1, two real noisy images captured by Samsung Galaxy S6 Edge and Google Pixel smartphones are chosen from Smartphone Image Denoising Dataset (SIDD) (Abdelhamed et al., 2018) and three $40 \times 40$ patches are chosen for illustration of noisy distribution. It is clear that real noise distribution is content dependent and noise in each patch has different statistical properties which can be non-Gaussian. Due to the violation of the AWGN assumption, the performance of

---

\*Corresponding author.

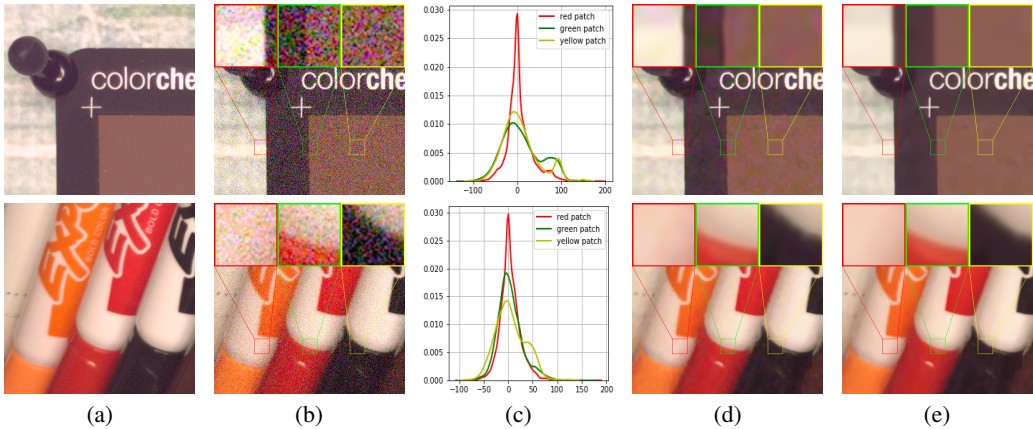

(a)           (b)           (c)           (d)           (e)

Figure 1: Two real noisy images. (a) Clean images. (b) Noisy Images. (c) Noisy distribution in red, green and yellow patches. (d) BM3D results (PSNR: 26.55 (top) and 29.41 (bottom)). (e) NN+BM3D (PSNR: 27.53 (top) and 30.05 (bottom)).

the Gaussian denoiser deteriorates significantly (Figure 1 (d)). Thus, it is crucial to characterize the noise distribution and adapt the noise models to the denoiser in real-world image denoising.

In recent years, deep learning based methods have achieved remarkable performance with careful architecture design, good training strategies, a large number of noisy and clean image pairs. However, there are two main drawbacks of these approaches from the perspective of practical applications. One is the high dependency on the quality and the size of the training dataset. Collecting such image pairs is time-consuming and requires much of human efforts, especially when the labeling needs deep domain knowledge such as medical or seismic images. The very recent deep learning methods including Noise2Noise (N2N) (Lehtinen et al., 2018), Noise2Void (N2V) (Krull et al., 2019a) and Noise2Self (N2S) (Batson & Royer, 2019) have relaxed the dataset requirement and can be trained on organized/un-organized noisy and noisy image pairs. Nevertheless, to guarantee the performance, these networks need to be pre-trained with a large number of images to cover sufficiently many local patterns, and thus they are not cost-effective. Therefore, to reduce the dependency of the training set, single-image based image denoising approaches deserved to be studied and have both practical and scientific value. It is worth mentioning that a recent unsupervised learning work (Ulyanov et al., 2018) uses a deep image prior to the general image recovery problem but its denoising results are inferior to some typical Gaussian denoisers, e.g., BM3D. The other drawback is the generalization ability of a trained network. When the noisy distribution is complicated and not contained in the training set, the results of the deep learning method can be deteriorated significantly, even worse than non-learning based methods. To alleviate this problem, some recent works are proposed by further consideration of noise estimation in the network design, e.g., Guo et al. (2019); Yue et al. (2019); Zhang et al. (2017). Despite their good performance in blind Gaussian denoising (Guo et al., 2019; Zhang et al., 2017) and real-world denoising problem (Yue et al., 2019), a large number of noisy and clean image pairs are needed and the generalization problem remains when the imaging system is complicated. Very recently, a single image based method has been proposed in (Quan et al., 2020) by developing a novel dropout technique for image denoising. Thus, unsupervised deep learning approaches with accurate noise models are important for solving real-world image denoising problems, yet current solutions are unsatisfactory. Such approach deserves to be studied and is a challenging problem as it needs a good combination of traditional methods and deep learning based methods such that the benefits of both methods are fully explored.

## 1.1 THE SUMMARY OF IDEAS AND CONTRIBUTIONS

Motivated by the above analysis, the goal of this paper is to propose an unsupervised deep learning method that boosts the performance of existing Gaussian denoisers when solving real-world image denoising problems. The basic idea is to find a latent image $\mathbf{z}$ associated with the input noisy image $\mathbf{y}$ such that $\mathbf{z}|\mathbf{x}$ satisfies the AWGN assumption, and thus we can obtain the clean image $\mathbf{x}$

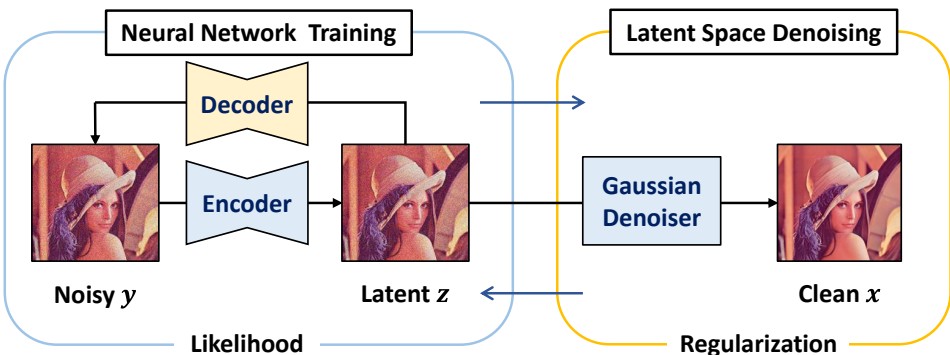

Figure 2: The workflow of our method. Middle arrows indicate the alternating optimization between network training and Gaussian denoising in latent space.

from **z** by any existing Gaussian denoiser. To find the appropriate latent representation, we propose a neural network (NN) based approach that builds up the mapping between the noisy image **y** and latent image **z** with an encoder-decoder structure. By applying the Gaussian denoiser in the latent space, we alternatively update the NN and the denoised image which does not need the other training samples. Figure 2 illustrates the workflow of the proposed approach.

Also, this idea can be formulated under the classical maximum a posterior (MAP) framework which consists of a likelihood term and a prior term. Building a proper likelihood term requires an accurate estimation of the noise distribution. Although the accurate noise distribution is difficult to get, an evidence lower bound (ELBO) can be analytically derived for approximating the likelihood from below using variational auto-encoder (VAE) (Kingma & Welling, 2013). This ELBO term gives the loss function for the encoder and decoder networks that are maps between noisy image and latent image. From the above derivation, we arrive at a model

$$\min_{\mathbf{x},\mathcal{F},\mathcal{G}} f(\mathbf{x}, \mathbf{y}, \mathcal{F}, \mathcal{G}) + R(\mathbf{x}), \tag{1}$$

where **x** is the clean image, **y** is the input noisy image, $\mathcal{F}, \mathcal{G}$ are decoder and encoder maps parameterized by NNs, $f$ is the loss from ELBO and $R(\mathbf{x})$ is the regularization term. Model (1) can be minimized by the alternative direction method of multiplier (ADMM) which alternatively updates networks and the clean image estimation **x**. Using Plug-and-Play technique (Venkatakrishnan et al., 2013), updating **x** can be replaced by any Gaussian denoiser. Thus, by fully exploiting the benefits of deep neural networks and classic denoising schemes, the real-world image denoising can be improved by a large margin as shown in Figure 1(e). More importantly, training the proposed networks only uses the noisy image itself and does not need any pre-training. In summary, we list our main contributions as follows.

- We propose an effective approach that combines the deep learning method with traditional methods for unsupervised image denoising. Thanks to the great expressive power of deep neural networks, the complex noise distribution is mapped into a latent space in which the AWGN assumption tends to hold, and thus better results are obtained by applying existing Gaussian denoiser for latent images.

- Instead of a heuristic loss design, the proposed NN approximates the likelihood in the classic Bayesian framework, which gives clear interpretations of each loss term. Meanwhile, compared to many existing deep learning methods, our model is only trained on a single image, which significantly reduces the burden of data collection.

- Extensive numerical experiments on real-world noisy image datasets have shown that the NN boosts the performance of the existing denoisers including NLM, BM3D and DnCNN. In particular, the results of NN+BM3D are competitive with some supervised deep learning approaches such as DnCNN+, FFDNet+, CBDNet.

## 2 RELATED WORK

There are numerous works in image denoising. Here, we review the works related to the non-learning and learning approaches for real-world image denoising.

### 2.1 NON-LEARNING BASED APPROACHES

Non-learning approaches are mainly based on the MAP framework that contains a data fidelity and a regularization term. Many works have been proposed to improve the regularize term, e.g., sparsity based methods (Rudin et al., 1992; Perona & Malik, 1990), low rank prior (Dong et al., 2012b; Gu et al., 2014) and non-local methods (Buades et al., 2005; Dabov et al., 2007a). Among these methods, BM3D (Dabov et al., 2007a) is one of the top methods. There are several other works related to the construction of data fidelity by modeling the complex noise distribution, e.g., Lebrun et al. (2015a); Nam et al. (2016); Xu et al. (2017); Zhu et al. (2016). The correlated Gaussian distribution (Lebrun et al., 2015a) and Mixture of Gaussian (Zhu et al., 2016; Nam et al., 2016) are used to approximate the unknown noise distribution. In Xu et al. (2017), different noise statistics are estimated in different channels without the consideration of the content dependent noise. Recently, Amini et al. (2020) proposed a Gaussianization method for gray scale OCT images. However, this method is not applicable for real-world image denoising tasks as natural images are colorful and the noise distribution is much more complicated than that in OCT images. Overall, due to the complexity of real-world noise, the performance of these approaches is unsatisfactory and needs to be improved.

### 2.2 LEARNING BASED APPROACHES

The learning based approaches can be classified into two groups: single-image based methods and dataset based methods. Typical single-image based approaches are sparse coding methods Aharon et al. (2006); Bao et al. (2015); Xu et al. (2018b). In Xu et al. (2018b), the noise in each channel is estimated and followed by a weighted sparse coding scheme. In recent years, as the appearance of real image denoising datasets including CC (Nam et al., 2016), PolyU (Xu et al., 2018a), DND (Plotz & Roth, 2017) and SIDD (Abdelhamed et al., 2018), deep neural networks including Guo et al. (2019); Yu et al. (2019); Zhang et al. (2017); Zhou et al. (2019); Yue et al. (2019) have shown promising results on these datasets. However, these networks require many noisy/clean training pairs which limit their practical applications especially when the labeling work needs domain experts. Recently, the deep learning approaches (Krull et al., 2019a;b; Batson & Royer, 2019; Laine et al., 2019; Lehtinen et al., 2018) are proposed and trained with organized or unorganized noisy image pairs. To guarantee a satisfactory performance, these methods still need many training pairs such that sufficiently many local patterns are covered. Compared to the above deep learning approaches, our method is a single image based method which does not need training samples or pre-training from other datasets.

## 3 OUR METHODOLOGY

This section starts with the derivation of our model and is followed by the detailed optimization techniques that incorporate any existing Gaussian denoiser.

### 3.1 THE MODEL FORMULATION

Let $\mathbf{y} \in \mathbb{R}^N$ be the noisy image where $N = \text{Height} \times \text{Width} \times 3$. The classic MAP framework aims at maximizing the posterior distribution $p(\mathbf{x}|\mathbf{y})$ formulated as the following optimization problem:

$$\max_{\mathbf{x}} \ln p(\mathbf{x}|\mathbf{y}) \propto \max_{\mathbf{x}} \ln p(\mathbf{y}|\mathbf{x}) + \ln p(\mathbf{x}) = \max_{\mathbf{x}} \ln p(\mathbf{y}|\mathbf{x}) - \lambda \mathcal{R}(\mathbf{x}). \tag{2}$$

The term $p(\mathbf{x})$ is the prior which represents the internal statistics of natural images. One common choice is $p(\mathbf{x}) \propto \exp(-\lambda \mathcal{R}(\mathbf{x}))$ where $\mathcal{R}(\mathbf{x})$ is a regularization function. The term $p(\mathbf{y}|\mathbf{x})$ is the likelihood that models the uncertainty of the observed image, i.e., $\mathbf{y} - \mathbf{x} \sim p(\mathbf{n})$ where $p(\mathbf{n})$ denotes the noise distribution. In practice, the noise distribution is complex and it is only possible to find an

approximation. Following the VAE approach (Kingma & Welling, 2013), the likelihood $p(\mathbf{y}|\mathbf{x})$ has a lower bound, i.e.,

$$\ln p(\mathbf{y}|\mathbf{x}) = \ln \int p(\mathbf{y}, \mathbf{z}|\mathbf{x}) d\mathbf{z} = \ln \int q(\mathbf{z}|\mathbf{y}) \frac{p(\mathbf{y}, \mathbf{z}|\mathbf{x})}{q(\mathbf{z}|\mathbf{y})} d\mathbf{z} \geq \int q(\mathbf{z}|\mathbf{y}) \ln \frac{p(\mathbf{y}, \mathbf{z}|\mathbf{x})}{q(\mathbf{z}|\mathbf{y})} d\mathbf{z}$$

$$= E_{q(\mathbf{z}|\mathbf{y})} \ln p(\mathbf{y}|\mathbf{z}, \mathbf{x}) - \mathrm{KL}(q(\mathbf{z}|\mathbf{y})||p(\mathbf{z}|\mathbf{x})) := \mathrm{ELBO},$$

where $\mathbf{z}$ is the latent image, $q$ is the distribution of the latent image $\mathbf{z}$ conditioned on the noisy image $\mathbf{y}$, KL denotes the Kullback–Leibler (KL) divergence and the ELBO stands for the evidence lower bound. In practice, we can construct a tractable ELBO with high expressive power, which motivates the usage of NNs.

The proposed NN consists of an encoder net $\mathcal{G}$ and a decoder net $\mathcal{F}$ that construct the mappings between the noisy image $\mathbf{y}$ and the latent image $\mathbf{z}$. Suppose the latent image $\mathbf{z}$ is the Gaussian corruption of the clean image $\mathbf{x}$ with strength $\sigma$, i.e., $\mathbf{z}|\mathbf{x} \sim \mathcal{N}(\mathbf{x}, \sigma^2 \mathbf{I})$ where $\mathcal{N}(0, \mathbf{I})$ is the standard Gaussian distribution. By choosing $p(\mathbf{y}|\mathbf{z}, \mathbf{x})$ and $q(\mathbf{z}|\mathbf{y})$ properly, the next proposition gives a closed form of the ELBO.

**Proposition 1** *Suppose $\mathbf{z}|\mathbf{x} \sim \mathcal{N}(\mathbf{x}, \sigma^2 \mathbf{I})$. Choosing $\mathbf{y}|\mathbf{z}, \mathbf{x} \sim \mathcal{N}(\mathcal{F}(\mathbf{z}), \mathbf{I})$ and $q(\mathbf{z}|\mathbf{y}) = \mathcal{N}(\mathcal{G}(\mathbf{y}), \mathbf{I})$ where $\mathcal{F}, \mathcal{G}$ are decoder and encoder respectively, the ELBO is equal to*

$$-\frac{1}{2} E_\epsilon ||\mathcal{F}(\mathcal{G}(\mathbf{y}) + \epsilon) - \mathbf{y}||^2 - \frac{1}{2\sigma^2} ||\mathcal{G}(\mathbf{y}) - \mathbf{x}||^2 + c, \tag{3}$$

*where $\epsilon \sim \mathcal{N}(0, \mathbf{I})$ and $c$ is a constant depends on the image size.*

The proof is given in appendix A.1. Let $\mathcal{F}$ and $\mathcal{G}$ be parameterized by $\theta_1$ and $\theta_2$ respectively, we replace the likelihood in (2) by (3) which derives our model:

$$E(\mathbf{x}, \theta_1, \theta_2) = \frac{1}{2} E_\epsilon ||\mathcal{F}_{\theta_1}(\mathcal{G}_{\theta_2}(\mathbf{y}) + \epsilon) - \mathbf{y}||^2 + \frac{1}{2\sigma^2} ||\mathcal{G}_{\theta_2}(\mathbf{y}) - \mathbf{x}||^2 + \lambda \mathcal{R}(\mathbf{x}). \tag{4}$$

There is an expectation term in (4), which can be estimated by the Monte Carlo method. Following the standard sampling approach in VAE (Kingma & Welling, 2013), we resample $\epsilon$ every time before backpropagation in our experiments.

**Remark 1** *In our model, the basic assumption is that the latent image $\mathbf{z}$ is a white Gaussian perturbations of clean image $\mathbf{x}$, i.e. $\mathbf{z}|\mathbf{x} \sim \mathcal{N}(\mathbf{x}, \sigma^2 \mathbf{I})$. Since $\mathbf{z} = \mathcal{G}_{\theta_2}(\mathbf{y})$, the second term in (4) can be seen as the transformed data fidelity. Moreover, the combination of those three terms can prevent the encoder from degenerating into zero mapping or identity mapping.*

### 3.2 MODEL OPTIMIZATION

Since the loss function (4) is non-convex and the regularization $\mathcal{R}(x)$ can be complicated, we introduce an auxiliary variable $\mathbf{p}$ and apply the ADMM scheme for solving (4). Concretely, define

$$f(\mathbf{x}, \theta_1, \theta_2) = \frac{1}{2} E_\epsilon ||\mathcal{F}_{\theta_1}(\mathcal{G}_{\theta_2}(\mathbf{y}) + \epsilon) - \mathbf{y}||^2 + \frac{1}{2\sigma^2} ||\mathcal{G}_{\theta_2}(\mathbf{y}) - \mathbf{x}||^2,$$

the loss function (4) is rewritten as

$$\min_{\mathbf{x}, \theta_1, \theta_2} f(\mathbf{x}, \theta_1, \theta_2) + \mathcal{R}(\mathbf{p}), \quad \text{s.t. } \mathbf{p} - \mathbf{x} = 0. \tag{5}$$

The augmented Lagrangian function of (5) is

$$\mathcal{L}_\rho(\mathbf{x}, \theta_1, \theta_2, \mathbf{p}) = f(\mathbf{x}, \theta_1, \theta_2) + \mathcal{R}(\mathbf{p}) + \frac{\rho}{2} ||\mathbf{x} - \mathbf{p} + \mathbf{q}/\rho||^2 - \frac{\rho}{2} ||\mathbf{q}/\rho||^2,$$

where $\mathbf{q}$ is the dual variable and $\rho > 0$ is a chosen constant. ADMM consists of the following iterates:

$$(\mathbf{x}^{k+1}, \theta_1^{k+1}, \theta_2^{k+1}) = \arg\min_{\mathbf{x}, \theta_1, \theta_2} \mathcal{L}_\rho(\mathbf{x}, \theta_1, \theta_2, \mathbf{p}^k, \mathbf{q}^k), \tag{6}$$

$$\mathbf{p}^{k+1} = \arg\min_{\mathbf{p}} \mathcal{L}_\rho(\mathbf{x}^{k+1}, \theta_1^{k+1}, \theta_2^{k+1}, \mathbf{p}, \mathbf{q}^k), \tag{7}$$

$$\mathbf{q}^{k+1} = \mathbf{q}^k + \rho(\mathbf{x}^{k+1} - \mathbf{p}^{k+1}).$$

Subproblem (6) is solved by alternating minimization, i.e., update the clean image estimation $\mathbf{x}$ and the network parameters $\theta_1$ and $\theta_2$ alternatively. More specifically, the networks $\mathcal{F}$ and $\mathcal{G}$ are updated by back propagation method with fixed $\mathbf{x}$. Then, $\theta_1$ and $\theta_2$ are fixed, $\mathbf{x}$ is updated by solving the minimization:

$$\min_{\mathbf{x}} \frac{1}{2\sigma^2} ||\mathbf{x} - \mathcal{G}_{\theta_2}(\mathbf{y})||^2 + \frac{\rho}{2} ||\mathbf{x} - \mathbf{p} + \mathbf{q}/\rho||^2. \tag{8}$$

The least square problem (8) has a closed form solution. Besides, subproblem (7) is equivalent to

$$\min_{\mathbf{p}} \mathcal{R}(\mathbf{p}) + \frac{\rho}{2} ||\mathbf{p} - \mathbf{x}^{k+1} - \mathbf{q}^k/\rho||^2.$$

Then $\mathbf{p}^{k+1} = \text{Prox}_{\rho \mathcal{R}}(\mathbf{x}^{k+1} + \mathbf{q}^k/\rho)$ where the proximal mapping is defined as $\text{Prox}_{\rho \mathcal{R}}(\mathbf{z}) = \arg \min_{\mathbf{x}} \mathcal{R}(\mathbf{z}) + \frac{\rho}{2} ||\mathbf{z} - \mathbf{x}||^2$. Motivated by this observation, the Plug-and-Play method (Venkatakrishnan et al., 2013) is proposed by generalizing this proximal mapping to any existing denoising scheme, i.e. $\mathbf{p}^{k+1} = \mathcal{T}(\mathbf{x}^{k+1} + \mathbf{q}^k/\rho)$. In our method, we choose the $\mathcal{T}$ as any existing Gaussian denoiser, e.g., NLM (Buades et al., 2005), BM3D (Dabov et al., 2007a). Overall, the detailed algorithm is given in Algorithm 1 where X denotes the Gaussian denoiser.

**Remark 2** *We initialized randomly for network parameters $\theta_1$ and $\theta_2$ without the usage of a pretrained model. In our method, $\mathbf{p}$ is initialized to $\mathbf{y}$, and $\mathbf{q}$ is initialized to 0 so that $\mathbf{x}$ is a linear combination of $\mathbf{y}$ and the latent space image at the early stage of this algorithm. Therefore, $\mathbf{x}$ is not far from $\mathbf{y}$ in the beginning, which will lead to a correct convergence for our algorithm.*

---

**Algorithm 1** The Denoising Algorithm NN+X.

---

**Input:** Noisy image $\mathbf{y}$, $\rho$, $\sigma$, $\eta$;
**Output:** Denoised image $\mathbf{x}$;
   Initial $\mathbf{x}^0 = \mathbf{y}$, $\mathbf{p}^0 = \mathbf{y}$, $\mathbf{q}^0 = 0$, and network parameters $\theta_1, \theta_2$.
   **for** $k = 0, 1, 2, 3, ..., M$ **do**
      **for** $i = 0, 1, 2, 3, ..., m$ **do**
         Sample the $\epsilon$ from standard Gaussian distribution in (6).
         Update $\theta_1^{k,i+1}, \theta_2^{k,i+1}$ by using backpropagation (BP) algorithm for (6).
         Update $\mathbf{x}^{k,i+1} = \left( \frac{1}{\sigma^2} \mathcal{G}_{\theta_2^{k,i+1}}(\mathbf{y}) + \rho \mathbf{p}^k - \mathbf{q}^k \right) / \left( \rho + \frac{1}{\sigma^2} \right)$.
      **end for**
      Set $\mathbf{x}^{k+1} = \mathbf{x}^{k,m}, \theta_1^{k+1} = \theta_1^{k,m}, \theta_2^{k+1} = \theta_2^{k,m}$.
      Update $\mathbf{p}^{k+1}$ by Gaussian Denoiser algorithm X to $\mathbf{x}^{k+1} + \mathbf{q}^k/\rho$.
      Update $\mathbf{q}^{k+1} = \mathbf{q}^k + \eta\rho(\mathbf{x}^{k+1} - \mathbf{p}^{k+1})$.
   **end for**
   **return** $\mathbf{x} = \mathbf{x}^N$.

---

## 4 EXPERIMENTS

We evaluate the performance of our method on real noisy images in this section. All experiments are evaluated in the sRGB space.

### 4.1 IMPLEMENTATION DETAILS

Both encoder network $\mathcal{G}$ and decoder network $\mathcal{F}$ are chosen as two standard 10 layers U-Nets (Ronneberger et al., 2015) implemented in Pytorch using Nvidia 1080TI or Nvidia 2080TI GPUs. ADAM algorithm (Kingma & Ba, 2014) is adopted to optimize the network parameters and the learning rate is set as 0.01. The number of epoch in network training is set as 500 and the parameters $\rho$, $\sigma$ and $\eta$ in ADMM are set as 1, 5 and 0.5 respectively. The noise level is estimated from Donoho & Johnstone (1994); Chen et al. (2015a). For an image with size $512 \times 512 \times 3$, our method needs about 15 minutes with a single Nvidia 2080TI GPU.

### 4.2 EXPERIMENTS ON REAL-WORLD NOISE

We combine the NN with three existing Gaussian denoiser methods, including two traditional methods (Non-local Mean (NLM) (Buades et al., 2005) and Block Matching and 3D Filtering

Table 1: Averaged PSNR and SSIM on CC, PolyU and FMDD.

| | CC | | PolyU | | FMDD | |
|---|---|---|---|---|---|---|
| | PSNR | SSIM | PSNR | SSIM | PSNR | SSIM |
| NLM | 33.47 | 0.8463 | 35.80 | 0.9063 | 30.81 | 0.8051 |
| VST+NLM | 36.05 | 0.9285 | 37.56 | 0.9545 | 31.34 | 0.7633 |
| NN+NLM | **37.31** | **0.9522** | **37.87** | **0.9606** | **31.96** | **0.8213** |
| BM3D | 35.19 | 0.8580 | 37.40 | 0.9526 | 29.70 | 0.7516 |
| VST+BM3D | 36.44 | 0.9396 | 38.15 | 0.9638 | 32.71 | 0.7922 |
| NN+BM3D | **38.26** | **0.9606** | **38.74** | **0.9681** | **33.91** | **0.8872** |
| DnCNN | 33.86 | 0.8635 | 36.08 | 0.9161 | 30.64 | 0.7514 |
| VST+DnCNN | 33.20 | 0.8452 | 35.66 | 0.9066 | 32.19 | 0.7728 |
| NN+DnCNN | **35.02** | **0.9069** | **36.67** | **0.9435** | **32.41** | **0.7798** |

Figure 3: Visual results and PNSR/SSIM of image from CC.

(BM3D) (Dabov et al., 2007b)) and one pre-trained deep learning method (DnCNN) (Zhang et al., 2017) from its official project website. We choose two nature real-world noisy image datasets named as CC (Nam et al., 2016), PolyU (Xu et al., 2018a), and one real fluorescence microscopy dataset named FMDD (Zhang et al., 2019) for testing the performance of our method in terms of PSNR and SSIM. Besides, the Variance Stabilizing Transform (VST) method (Makitalo & Foi, 2012), a traditional noisy transformation method, is chosen for comparison. We employ the noise estimation method (Foi et al., 2008) to estimate the Poisson-Gaussian noise parameters for the VST method in FMDD. In CC and PolyU, we evaluate different Poisson noise parameters (peak value, range from 10 to 10000) and select the best for whole datasets. There are 15 and 100 images in CC (Nam et al., 2016) and PolyU (Xu et al., 2018a) datasets with the same cropped regions in their original papers. In FMDD, we evaluate the performance on the mixed test set with raw images which is the same setting as in Zhang et al. (2019).

**Denoising results.** The denoising results are evaluated using PSNR and SSIM, with built-in functions in Python skimage package. The results are listed in Table 1 from which we have the following observations: (a) Our model improves the performance of both three Gaussian denoiser by a large margin on both two datasets. On average, the PSNR/SSIM values are increased by 2.35/0.0588, 2.87/0.0844 and 1.173/0.0331 for NLM, BM3D and DnCNN respectively, see Figure 3 for one visual example from CC. (b) Comparing with the VST method, the existing Gaussian Denoisers benefit more from the proposed method.

## 5 DISCUSSION

**Validation of AWGN assumption of latent images.** Let $\mathbf{x}, \mathbf{y}, \mathbf{z}$ be clean, noisy and latent images. Denote $\mathbf{n}_1 = \mathbf{y} - \mathbf{x}$ and $\mathbf{n}_2 = \mathbf{z} - \mathbf{x}$ are the noise in image space and latent space respectively. We visualize the distribution of $\mathbf{n}_1$ and $\mathbf{n}_2$ in Figure 4 using two images in Figure 1. It is obvious that the noise distribution in latent space is more similar to a white Gaussian than in image space.

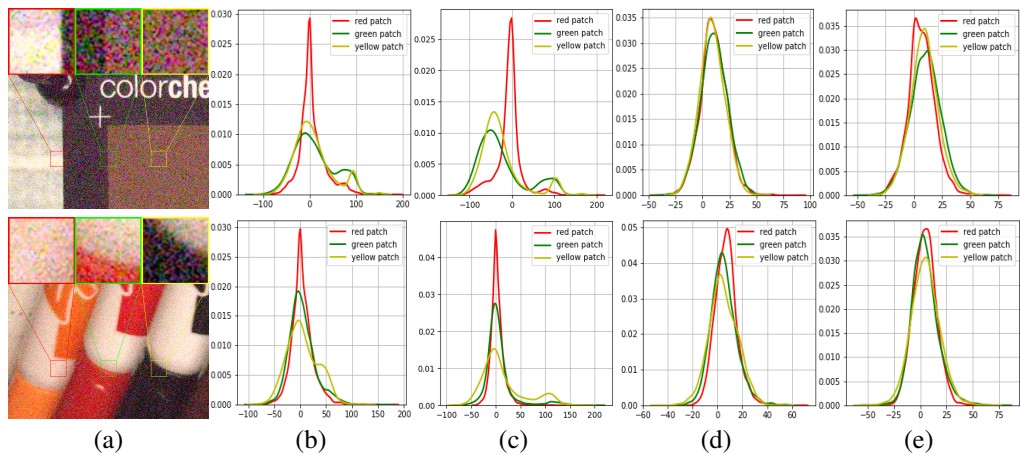

Figure 4: (a) Noisy images with three selected patches. (b) Noisy distribution in image space. (c) Noisy distribution after VST transformation. (d-e) Noisy distribution in latent space by NN+NLM and NN+BM3D respectively.

Table 2: Comparison with supervised/unsupervised methods on CC.

| Metric | Supervised methods | | | Unsupervised methods | | | NN+BM3D |
|---|---|---|---|---|---|---|---|
| | DnCNN+ | FFDNet+ | CBDNet | N2N | NAC | DIP* | |
| PSNR | 35.40 | 37.63 | 36.44 | 35.32 | 36.59 | 37.72 | **38.26** |
| SSIM | 0.9115 | 0.9555 | 0.9460 | 0.9160 | 0.9502 | 0.9531 | **0.9606** |

Moreover, define $a_i = \text{Mean}(\mathbf{n}_i)$ and $\sigma_i^2 = \text{Var}(\mathbf{n}_i)$ for $i = 1, 2$ where Mean, Var are the mean and variance operators respectively. To quantify the distance of between $\mathbf{n}_i, i = 1, 2$ and white Gaussian distribution, we calculate the KL divergence between $\mathbf{n}_i$ and $\mathcal{N}(a_i, \sigma_i^2 \mathbf{I})$ on CC dataset. In image space, the KL divergence is 0.4701, while in latent space the KL divergence reduces to 0.3821, 0.2868 and 0.3636 for NN+NLM, NN+BM3D and NN+DnCNN respectively. We note the noise distribution in latent space is closer to the Gaussian distribution than its in image space.

**Comparison with supervised/unsupervised methods.** Three blind image denoising networks including DnCNN+(Zhang et al., 2017), FFDNet+ (Zhang et al., 2018), CBDNet (Guo et al., 2019) and three unsupervised approaches including Noise2Noise (N2N) (Lehtinen et al., 2018), Deep Image Prior (DIP*) (Ulyanov et al., 2018) (3000 iterations for 5 times average) and Noise-As-Clean (NAC) (Xu et al., 2019) are chosen for comparison on CC datasets. FFDNet+ is a multi-scale extension of FFDNet (Zhang et al., 2018) and DnCNN+ is a color version of DnCNN and fine tuned with the FFDNet+ results[1]. The denoising results are listed in Table 2. It is shown that NN+BM3D is better than deep neural networks trained on a dataset and other unsupervised deep learning methods.

**The capacity of decoder.** The proposed model (4) has a trivial global minimum when assuming (a) the decoder is a constant mapping, i.e. $\mathcal{F}(\mathbf{z}) = \mathbf{y}$ for all $\mathbf{z}$; (b) $\mathcal{G}(\mathbf{y}) = \mathbf{x} = 0$. We test the capacity of the decoder used in our model by fitting the map between the constant image and the noisy image, and the training loss versus iteration is reported in Figure 5 (a). Moreover, the average training loss on PolyU dataset is 294.56, which shows that our decoder is not large enough and the trivial minimum does not exist, see more explanation in appendix A.2.

**Training stability.** Four classic images including Kodim03 (red), Kodim02 (green), Lena (blue) and Peppers (yellow) with noise level $\sigma = 30$ are used for testing the training stability of our method. The PSNR value versus iteration number is reported in Figure 5 (b) and it shows that the PSNR keeps increasing and is stable after a certain number of iterations.

**Latent space evolution.** One image in the introduction from SIDD dataset is used to show the latent image evolution in our method. In particular, we evaluate NN+BM3D for 20 iterations and show

---

[1]The results of DnCNN+, FFDNet+ and CBDNet are from (Hou et al., 2019) and the results of N2N and NAC are from (Xu et al., 2019).

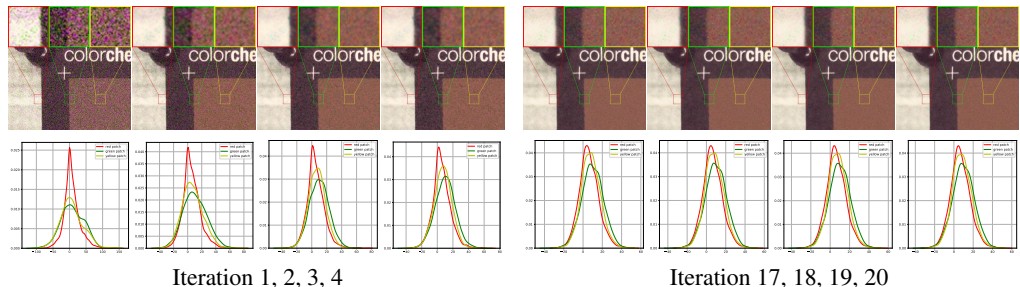

Figure 5: (a) is the training losses of our decoder using constant input for five images from PolyU. (b) show PSNR value versus the number of iterations in NN+BM3D. (c), (d) and (e) are results for different hyperparameters $\rho$, $\sigma$ and $\eta$ respectively.

Iteration 1, 2, 3, 4          Iteration 17, 18, 19, 20

Figure 6: Evaluation of the latent representation in Algorithm 1 with NN+BM3D.

8 latent images and their patch noise distribution evolution in Figure 6. It is shown that the noise distribution gradually changes from non-Gaussian to Gaussian.

**Hyperparameter sensitivity.** The sensitivity of hyperparameters are given in Figure 5 (c), (d) and (e), where we change one of $\rho$, $\sigma$ and $\eta$ each time and fix the others as $\rho = 1$, $\sigma = 5$ and $\eta = 0.5$. It can be observed that our method is not sensitive to different hyperparameters.

## 6    CONCLUSION

In this paper, we propose a NN based method that maps the complex noisy distribution in real-world images into a latent space in which the AWGN assumption holds. Combined with any existing Gaussian denoising approaches, it improves the denoising results by a large margin. More importantly, this method does not require any training sample except the input noisy image itself. Extensive results validate the rationale of the proposed network training scheme and show the advantages of our method compared with existing representative approaches including learning and non-learning based methods.

## ACKNOWLEDGEMENTS

This work is supported by the National Natural Science Foundation of China (11901338, 12071244), Technology and Innovation Major Project of the Ministry of Science and Technology of China under Grant 2020AAA0108400 and 2020AAA0108403 and Tsinghua University Initiative Scientific Research Program.

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

## A    PROOF OF PROPOSITIONS 1

We show the derivation of Proposition 1 and give one theoretical explanation of the capacity of decoder.

### A.1    PROPOSITION 1

The ELBO in Proposition 1 is defined as $E_{q(\mathbf{z}|\mathbf{y})} \ln p(\mathbf{y}|\mathbf{z}, \mathbf{x}) - \mathrm{KL}(q(\mathbf{z}|\mathbf{y})||p(\mathbf{z}|\mathbf{x}))$. Since $\mathbf{z}|\mathbf{x} \sim \mathcal{N}(\mathbf{x}, \sigma^2 \mathbf{I})$, $\mathbf{y}|\mathbf{z}, \mathbf{x} \sim \mathcal{N}(\mathcal{F}(\mathbf{z}), \mathbf{I})$ and $q(\mathbf{z}|\mathbf{y}) = \mathcal{N}(\mathcal{G}(\mathbf{y}), \mathbf{I})$. Assume the image size of $\mathbf{x}$ is $m \times n \times 1$ for gray scale image, then the KL divergence of two Gaussian distribution is

$$
\begin{aligned}
\mathrm{KL}(q(\mathbf{z}|\mathbf{y})||p(\mathbf{z}|\mathbf{x})) &= E_{q(\mathbf{z}|\mathbf{y})} \ln q(\mathbf{z}|\mathbf{y}) - \ln p(\mathbf{z}|\mathbf{x}) \\
&= \frac{mn}{2} \ln \sigma^2 + \frac{1}{2} E_{q(\mathbf{z}|\mathbf{y})} \left( -||\mathbf{z} - \mathcal{G}(\mathbf{y})||^2 + \frac{1}{\sigma^2} ||\mathbf{z} - \mathbf{x}||^2 \right) \\
&= \frac{mn}{2} \ln \sigma^2 - \frac{mn}{2} + \frac{1}{2\sigma^2} E_{q(\mathbf{z}|\mathbf{y})} ||\mathbf{z} - \mathbf{x}||^2 \\
&= \frac{mn}{2} \ln \sigma^2 - \frac{mn}{2} + \frac{1}{2\sigma^2} E_{q(\mathbf{z}|\mathbf{y})} ||\mathbf{z} - \mathcal{G}(\mathbf{y}) + \mathcal{G}(\mathbf{y}) - \mathbf{x}||^2 \\
&= \frac{mn}{2} \ln \sigma^2 - \frac{mn}{2} + \frac{1}{2\sigma^2} \left( mn + ||\mathcal{G}(\mathbf{y}) - \mathbf{x}||^2 \right) \\
&= \frac{mn}{2} \left( \ln \sigma^2 - 1 + \frac{1}{\sigma^2} \right) + \frac{1}{\sigma^2} ||\mathcal{G}(\mathbf{y}) - \mathbf{x}||^2.
\end{aligned}
$$

Using the reparameterization in (Kingma & Welling, 2013), $\mathbf{z}|\mathbf{y} = \mathcal{G}(\mathbf{y}) + \epsilon$ where $\epsilon \sim \mathcal{N}(0, \mathbf{I})$, the expectation term is

$$
\begin{aligned}
E_{q(\mathbf{z}|\mathbf{y})} \ln p(\mathbf{y}|\mathbf{z}, \mathbf{x}) &= -\frac{mn}{2} \ln 2\pi - \frac{1}{2} E_{q(\mathbf{z}|\mathbf{y})} ||\mathcal{F}(\mathbf{z}) - \mathbf{y}||^2 \\
&= -\frac{mn}{2} \ln 2\pi - \frac{1}{2} E_{\epsilon} ||\mathcal{F}(\mathcal{G}(\mathbf{y}) + \epsilon) - \mathbf{y}||^2
\end{aligned}
$$

Then the ELBO $= E_{q(\mathbf{z}|\mathbf{y})} \ln p(\mathbf{y}|\mathbf{z}, \mathbf{x}) - \mathrm{KL}(q(\mathbf{z}|\mathbf{y})||p(\mathbf{z}|\mathbf{x}))$ is

$$
-\frac{1}{2} E_{\epsilon} ||\mathcal{F}(\mathcal{G}(\mathbf{y}) + \epsilon) - \mathbf{y}||^2 - \frac{1}{2\sigma^2} ||\mathcal{G}(\mathbf{y}) - \mathbf{x}||^2 - \frac{mn}{2} \left( \ln 2\pi + \ln \sigma^2 - 1 + \frac{1}{\sigma^2} \right),
$$

and for RGB images the constant term becomes $-\frac{3mn}{2} \left( \ln 2\pi + \ln \sigma^2 - 1 + \frac{1}{\sigma^2} \right)$, thus Proposition 1 holds.

### A.2    THE CAPACITY OF DECODER

Here we give an explanation regarding to capacity of decoder in the following proposition.

**Proposition 2** *Suppose $\mathbf{x} \in \mathbb{R}^{m \times n \times c}$ is a constant image, i.e.*

$$
\mathbf{x}_{i,j,k} = \mathbf{x}_{i',j',k}, \quad \forall i, i' \in \{1, 2, ..., m\}, j, j' \in \{1, 2, ..., n\}, k \in \{1, 2, ..., c\},
$$

*and $\mathcal{F}$ is a deep neural network that composted with convolution layers (with reflect/replicate padding), down-sampling layers (with max/average pooling), and up-sampling layer (with bilinear/nearest point interpolation) then $\mathcal{F}(\mathbf{x})$ is constant.*

To simplify notations, we denote convolution layers as $C$, down-sampling layers as $D$, and up-sampling layer as $U$, then we only need to show for constant input $\mathbf{x} \in \mathbb{R}^{m \times n \times c}$, $C(\mathbf{x})$, $D(\mathbf{x})$ and

$U(\mathbf{x})$ are constants. Assume $C(x) = \tilde{\mathbf{x}} \in \mathbb{R}^{m' \times n' \times c'}$, we need to show $\tilde{\mathbf{x}}_{i,j,k} = \tilde{\mathbf{x}}_{i',j',k}$. From the dimension of $\tilde{\mathbf{x}}$, there are $c'$ convolution kernels denote as $\mathcal{K}^i \in \mathbb{R}^{p_i \times q_i \times c}, i = 1, 2, ..., c'$, then

$$\tilde{\mathbf{x}}_{i,j,k} = \sum_{l_1,l_2,l_3} \mathcal{K}^k_{l_1,l_2,l_3} \mathbf{x}_{i-\frac{p_k-1}{2}+l_1, j-\frac{q_k-1}{2}+l_2, l_3}$$

$$= \sum_{l_1,l_2,l_3} \mathcal{K}^k_{l_1,l_2,l_3} \mathbf{x}_{i'-\frac{p_k-1}{2}+l_1, j'-\frac{q_k-1}{2}+l_2, l_3}$$

$$= \tilde{\mathbf{x}}_{i',j',k},$$

so $C(x)$ is constant. For down-sampling layer, since $D(\mathbf{x})_{i,j,k} = \max\{\mathbf{x}_{i,j,k} | \forall i, j \in \mathcal{R}\}$ for max pooling and $D(\mathbf{x})_{i,j,k} = \frac{1}{|\mathcal{R}|} \sum_{i,j \in \mathcal{R}} \mathbf{x}_{i,j,k}$ for average pooling, where $\mathcal{R}$ denote support of pooling kernel. Since $\mathbf{x}$ is constant, then it is clear from above expression that $P(\mathbf{x})$ is constant. Finally, for up-sampling layer, since we assume bilinear/nearest point interpolation, then $U(\mathbf{x})_{i,j,k}$ depends on the convex combination of $\{\mathbf{x}_{i,j,k} | \forall i, j \in \mathcal{S}\}$, where $\mathcal{S}$ denote the corresponding up-sampling region of $U(\mathbf{x})_{i,j,k}$. So if $\mathbf{x}$ is constant then the convex combination of $\mathcal{S}$ is constant, then we have $U(\mathbf{x})$ is constant. Thus proposition 2 holds.

The above proposition shows that the latent image can not be a constant image and thus the regularization takes effects. Besides, the minimization problem is a non-convex problem which depends on the initialization. In our experiments, we initialize $\mathbf{x}$ to be the input noisy image $\mathbf{y}$ and it empirically converges to a reasonable good denoised image.

Table 3: Average PSNR/SSIM of denoised results on Set9 and BSD 68.

| Datasets | Methods | $\sigma = 25$ | $\sigma = 50$ | $\sigma = 75$ | $\sigma = 100$ |
|---|---|---|---|---|---|
| BSD68 | NLM | 25.40/0.6281 | 21.27/0.4120 | 19.39/0.3092 | 18.52/0.2597 |
| | NN+NLM | **27.19/0.7493** | **23.68/0.5663** | **21.48/0.4734** | **19.65/0.4329** |
| | BM3D | **28.33**/0.8029 | 24.86/0.6698 | 22.63/0.5808 | 20.77/0.5167 |
| | NN+BM3D | 28.32/**0.8040** | **24.91/0.6776** | **22.75/0.5896** | **20.90/0.5241** |
| Set9 | NLM | 30.11/0.7876 | 25.98/0.6615 | 22.59/0.5620 | 20.35/0.4930 |
| | NN+NLM | **30.27/0.7930** | **26.29/0.6761** | **22.77/0.5728** | **20.45/0.4959** |
| | BM3D | 31.47/0.8406 | 27.85/0.7551 | 24.64/0.6834 | 21.96/0.6198 |
| | NN+BM3D | **31.48/0.8415** | **27.98/0.7618** | **24.79/0.6937** | **22.06/0.6285** |

## B  EXPERIMENTS ON SYNTHETIC NOISE

**Additive white Gaussian noise.** Even though our method is designed for the real-world image denoising, we evaluate the performance of this method on synthetic noises. We test two different Gaussian denoising methods, NLM and BM3D. Two datasets are used, one is the Set 9 (Ulyanov et al., 2018) with 9 classic color images and the other is BSD 68 (Krull et al., 2019a) with 68 grayscale images. We test four noisy levels: $\sigma = 25, 50, 75, 100$, and all noisy images are quantized into 8-bits for simulating the common JPG images. The results are given in Table 3 we find even though the noise distribution is Gaussian, two denoising methods can benefit from our approach more or less.

**Poisson noise.** For noise distribution which is significantly different from Gaussian, we test the synthetic Poisson noise on CBSD 68 dataset with 68 color images by combining our method with VST transformation. The results are given in Table 4, all noisy images are quantized into 8-bits. We find that even though VST is a reliable transformation for the pure Poisson noise case, the neural networks are still helpful for improving the VST method. For DnCNN, we find it sensitive to Poisson noise, for Peak value equals to 5, the result of VST+DnCNN is 17.23/0.3767 and our method (NN+VST+DnCNN) improves 0.60/0.0629 for it on average.

Table 4: Average PSNR/SSIM of Poisson denoising results on CBSD 68.

| Peak Value | 5 | 7 | 9 | 20 |
|---|---|---|---|---|
| VST+BM3D | 23.85/0.6792 | 24.86/0.7201 | 25.66/0.7490 | 27.99/0.8252 |
| NN+VST+BM3D | **24.07/0.6951** | **25.06/0.7340** | **25.82/0.7610** | **28.01/0.8313** |

Table 5: Averaged PSNR/SSIM on CC, PolyU, DND, SIDD.

| Datasets | Methods | | | | |
|---|---|---|---|---|---|
| CC | WNNM | NCSR | TNRD | DnCNN | BM3D |
| | 35.77/0.9381 | 33.46/0.8591 | 36.61/0.9463 | 33.86/0.8635 | 35.19/0.8580 |
| | MCWNNM | TWSC | NC | NI | NN+BM3D |
| | 37.71/0.9542 | 37.81/0.9586 | 36.43/0.9364 | 35.49/0.9126 | **38.26/0.9606** |
| PolyU | WNNM | NCSR | TNRD | DnCNN | BM3D |
| | 36.59/0.9247 | 36.40/0.9290 | 38.17/0.9640 | 36.08/0.9161 | 37.40/0.9526 |
| | MCWNNM | TWSC | NC | NI | NN+BM3D |
| | 38.51/0.9671 | 38.60/**0.9685** | 36.92/0.9449 | 37.77/0.9570 | **38.74**/0.9681 |
| DND | WNNM | NCSR | TNRD | DnCNN | BM3D |
| | 34.67/0.8646 | 34.05/0.8351 | 33.65/0.8306 | 32.4296/0.79 | 34.51/0.8507 |
| | MCWNNM | TWSC | NC | NI | NN+BM3D |
| | 37.379/0.9294 | **37.96**/0.9416 | 35.434/0.8841 | 35.1125/0.8778 | 37.10/**0.9441** |
| SIDD | WNNM | NCSR | TNRD | DnCNN | BM3D |
| | 25.78/0.809 | – | 24.73/0.643 | 23.66/0.583 | 25.65/0.685 |
| | MCWNNM | TWSC | NC | NI | NN+BM3D |
| | – | – | – | – | **33.18/0.895** |

## C  EXPERIMENTS ON REAL WORLD NOISE

In this section, We compare the performance of NN+BM3D with other denoising method on four real-world noisy image datasets named as CC (Nam et al., 2016), PolyU (Xu et al., 2018a), DND (Plotz & Roth, 2017) and SIDD (Abdelhamed et al., 2018). we make a comprehensive comparison study of our method with many existing representative image denoising methods, including Weighted Nuclear Norm Minimization (WNNM) (Gu et al., 2014), Nonlocally Centralized Sparse Representation (NCSR) (Dong et al., 2012a), Trainable Nonlinear Reactive Diffusion (TNRD) (Chen et al., 2015b), DnCNN (Zhang et al., 2017), Multi-channel Weighted Nuclear Norm Minimization (MCWNNM) (Xu et al., 2017), Trilateral Weighted Sparse Coding (TWSC) (Xu et al., 2018b), "Noise Clinic" (NC) method (Lebrun et al., 2015a) and a commercial software Neat Image (NI) (ABSoft, 2017). See Table 5 for the averaged PSNR/SSIM results. The results of comparison methods in CC and PolyU datasets are from (Xu et al., 2018a) and their results of DND and SIDD datasets are from their official project website.

Table 6: Comparison with supervised methods.

| Datasets | DnCNN+ | FFDNet+ | CBDNet | NN+BM3D |
|---|---|---|---|---|
| DND | 37.90/0.943 | 37.61/0.9415 | **38.05**/0.9421 | 37.10/**0.9441** |
| SIDD | – | – | **33.28**/0.868 | 33.10/**0.895** |

Table 7: Comparison with supervised/unsupervised methods on FMDD.

| Metric | Unsupervised Methods | | | | Supervised Method | |
|---|---|---|---|---|---|---|
| | VST+BM3D | PURE-LET | TWSC | NN+BM3D | DnCNN | N2N |
| PSNR | 32.71 | 31.95 | 31.64 | **33.91** | 34.88 | **35.40** |
| SSIM | 0.7922 | 0.7664 | 0.7787 | **0.8872** | 0.9063 | **0.9187** |

In addition, we compare the performance of NN+BM3D with Three top blind image denoising networks including DnCNN+(Zhang et al., 2017), FFDNet+ (Zhang et al., 2018) and CBDNet (Guo et al., 2019). According to the results in DND and SIDD benchmarks, the denoising results are listed in Table 6. The symbol "-" is used when the result of the corresponding method is not reported.

We also compare the performance of NN+BM3D with supervised/unsupervised methods in FMDD dataset (Zhang et al., 2019). We compared with three unsupervised methods named as VST+BM3D (Makitalo & Foi, 2012), PURE-LET (Luisier et al., 2010) and TWSC (Xu et al., 2018b), two supervised methods DnCNN (Zhang et al., 2017) and N2N (Lehtinen et al., 2018) which are retrained on the FMDD training set. The denoising results are listed in Table 7. Although

Table 8: Averaged PSNR/SSIM of different architectures of encoder/decoder on CC dataset.

| UNet D4 | UNet D3 | UNet D2 |
|---|---|---|
| 38.32/0.9609 | 37.89/0.9573 | 37.98/0.9581 |
| UNet wo Skip D4 | UNet wo Skip D3 | UNet wo Skip D2 |
| 36.65/0.9499 | 37.37/0.9543 | 37.69/0.9555 |
| R(C(64)-C(64)) | R(C(32)-C(64)-C(32)) | R(C(64)-C(128)-C(64)) |
| 38.34/0.9610 | 38.34/0.9609 | 38.35/0.9609 |

Table 9: Averaged PSNR/SSIM of NN+FFDNet on CC dataset.

| Method | DnCNN | NN+DnCNN | FFDNet | FFDNet+ | NN+FFDNet |
|---|---|---|---|---|---|
| PSNR | 33.86 | 35.02 | 34.63 | 37.63 | **38.00** |
| SSIM | 0.8635 | 0.9069 | 0.8551 | 0.9555 | **0.9574** |

NN+BM3D is not as good as DnCNN and N2N which are specially refined in the dataset, it is better than all unsupervised method in terms of PSNR and SSIM.

## D   ABLATION EXPERIMENTS

**Different architectures of NN.** Here we show the results of different choose of the architectures of neural network. We evaluate our method with UNets of different downsample scales (denote by D), with or without skip connection (denote by wo skip), and different CNNs, denote C(n) as Convolution(n)-BatchNorm-ReLU, R(model)(x) = x + model(x). The results on CC dataset are given in Table 8

**NN+FFDNet.** We replaced DnCNN with FFDNet, and the result on CC is given in Table 9, it is shown that FFDNet benefits more significantly from our method and is better than the results of its multi-scale version FFDNet+.

## E   VISUAL RESULTS ON REAL WORLD NOISE

We show visual results on real world noisy images in E.1 and real fluorescence microscopy images in E.2. Moreover, ten real noisy images are captured by consumer cameras with ISO=3200 or 320, similar to the CC dataset, we crop a $512 \times 512$ region in each image to evaluate the performance NN+BM3D, see E.3.

### E.1   VISUAL EXAMPLE OF CC, POLYU

We show visual results of ten noisy images from CC (Nam et al., 2016) and PolyU (Xu et al., 2018a) datasets. BM3D (Dabov et al., 2007a), DnCNN (Zhang et al., 2017), NC (Lebrun et al., 2015b), MCWNNM (Xu et al., 2017) and TWSC (Xu et al., 2018b) are evaluated for comparison. See CC's results in page 16, 17 and PolyU's results in page 18, 19.

### E.2   VISUAL EXAMPLE OF FMDD

Two images from FMDD Zhang et al. (2019) datasets are evaluated for visual comparisons. We compared our approach with VST method (Makitalo & Foi, 2012), See page 20.

### E.3   VISUAL EXAMPLES OF REAL IMAGE

Ten real noisy images are evaluated for visual comparisons. The results of three traditional methods (BM3D (Dabov et al., 2007a), MCWNNM (Xu et al., 2017) and NC (Lebrun et al., 2015b)) and three deep learning methods (VDN (Yue et al., 2019), DnCNN (Zhang et al., 2017) and FFDNet (Zhang et al., 2018)) are shown here. See Figure 10 in page 21, 22, 23.

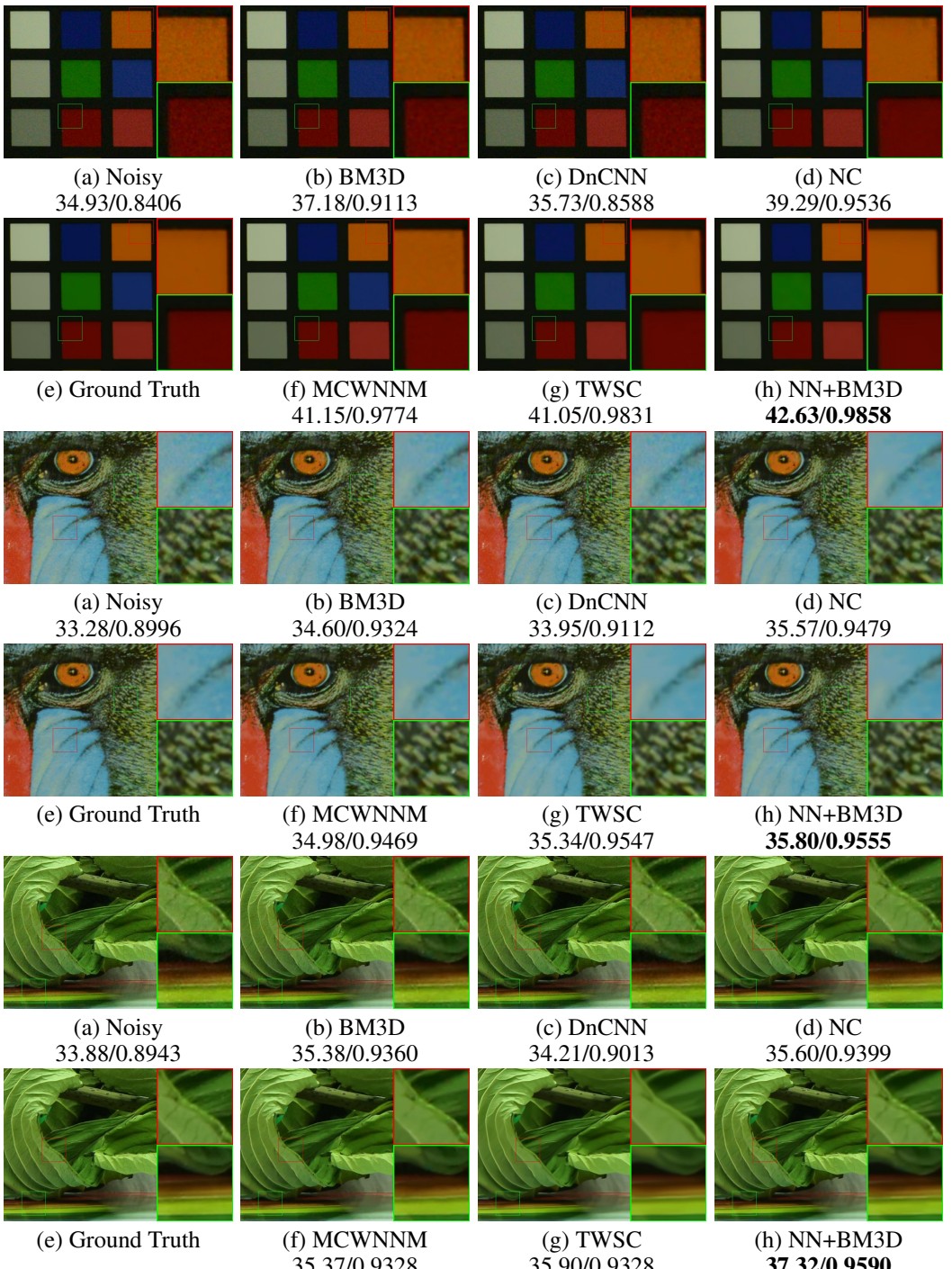

(a) Noisy
34.93/0.8406

(b) BM3D
37.18/0.9113

(c) DnCNN
35.73/0.8588

(d) NC
39.29/0.9536

(e) Ground Truth

(f) MCWNNM
41.15/0.9774

(g) TWSC
41.05/0.9831

(h) NN+BM3D
**42.63/0.9858**

(a) Noisy
33.28/0.8996

(b) BM3D
34.60/0.9324

(c) DnCNN
33.95/0.9112

(d) NC
35.57/0.9479

(e) Ground Truth

(f) MCWNNM
34.98/0.9469

(g) TWSC
35.34/0.9547

(h) NN+BM3D
**35.80/0.9555**

(a) Noisy
33.88/0.8943

(b) BM3D
35.38/0.9360

(c) DnCNN
34.21/0.9013

(d) NC
35.60/0.9399

(e) Ground Truth

(f) MCWNNM
35.37/0.9328

(g) TWSC
35.90/0.9328

(h) NN+BM3D
**37.32/0.9590**

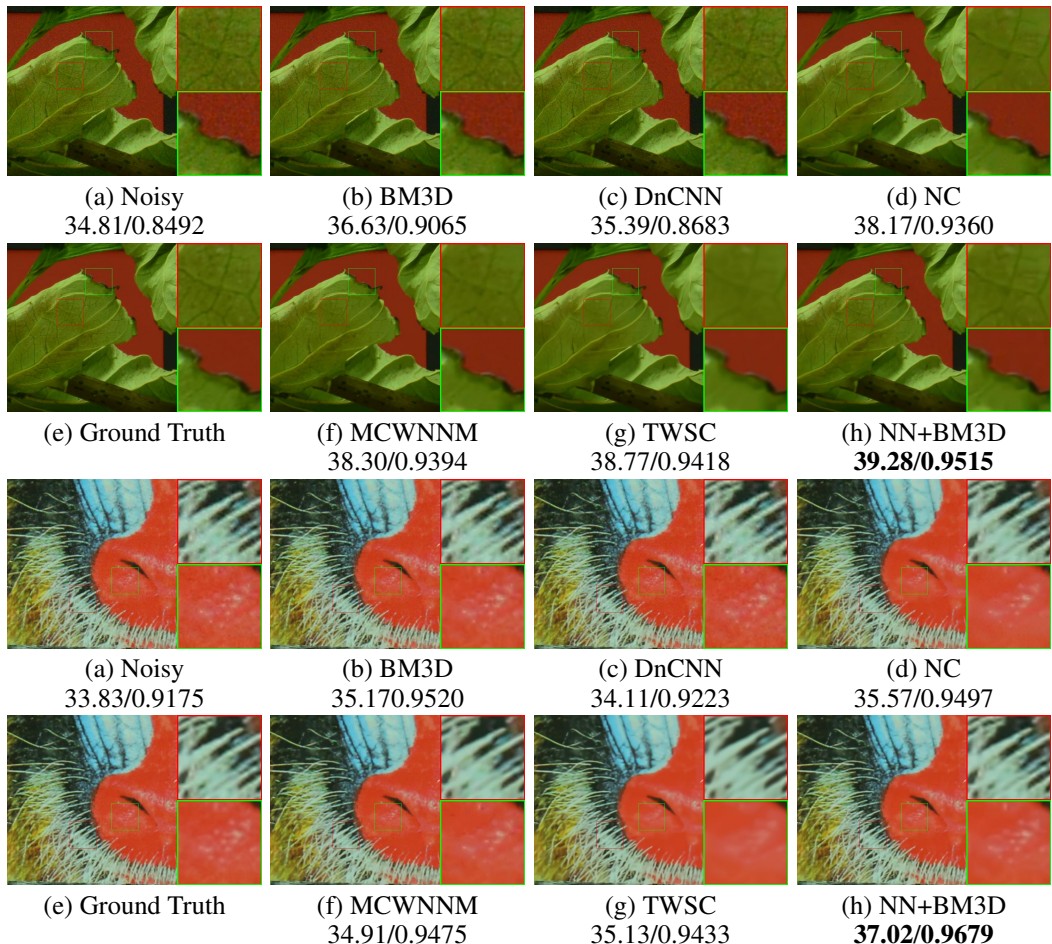

Figure 7: Visual results and PNSR/SSIM of five noisy images from CC.

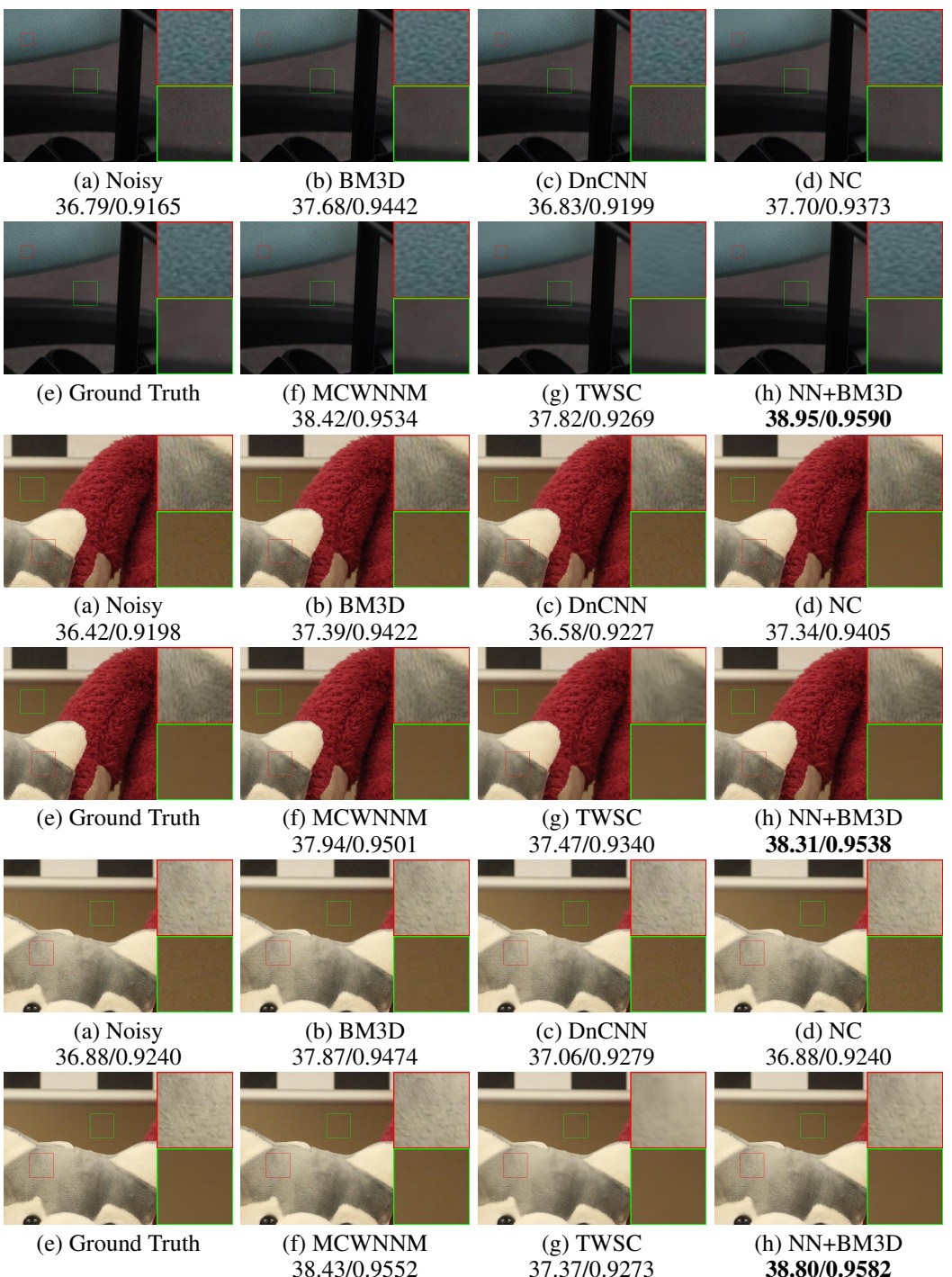

(a) Noisy
36.79/0.9165

(b) BM3D
37.68/0.9442

(c) DnCNN
36.83/0.9199

(d) NC
37.70/0.9373

(e) Ground Truth

(f) MCWNNM
38.42/0.9534

(g) TWSC
37.82/0.9269

(h) NN+BM3D
**38.95/0.9590**

(a) Noisy
36.42/0.9198

(b) BM3D
37.39/0.9422

(c) DnCNN
36.58/0.9227

(d) NC
37.34/0.9405

(e) Ground Truth

(f) MCWNNM
37.94/0.9501

(g) TWSC
37.47/0.9340

(h) NN+BM3D
**38.31/0.9538**

(a) Noisy
36.88/0.9240

(b) BM3D
37.87/0.9474

(c) DnCNN
37.06/0.9279

(d) NC
36.88/0.9240

(e) Ground Truth

(f) MCWNNM
38.43/0.9552

(g) TWSC
37.37/0.9273

(h) NN+BM3D
**38.80/0.9582**

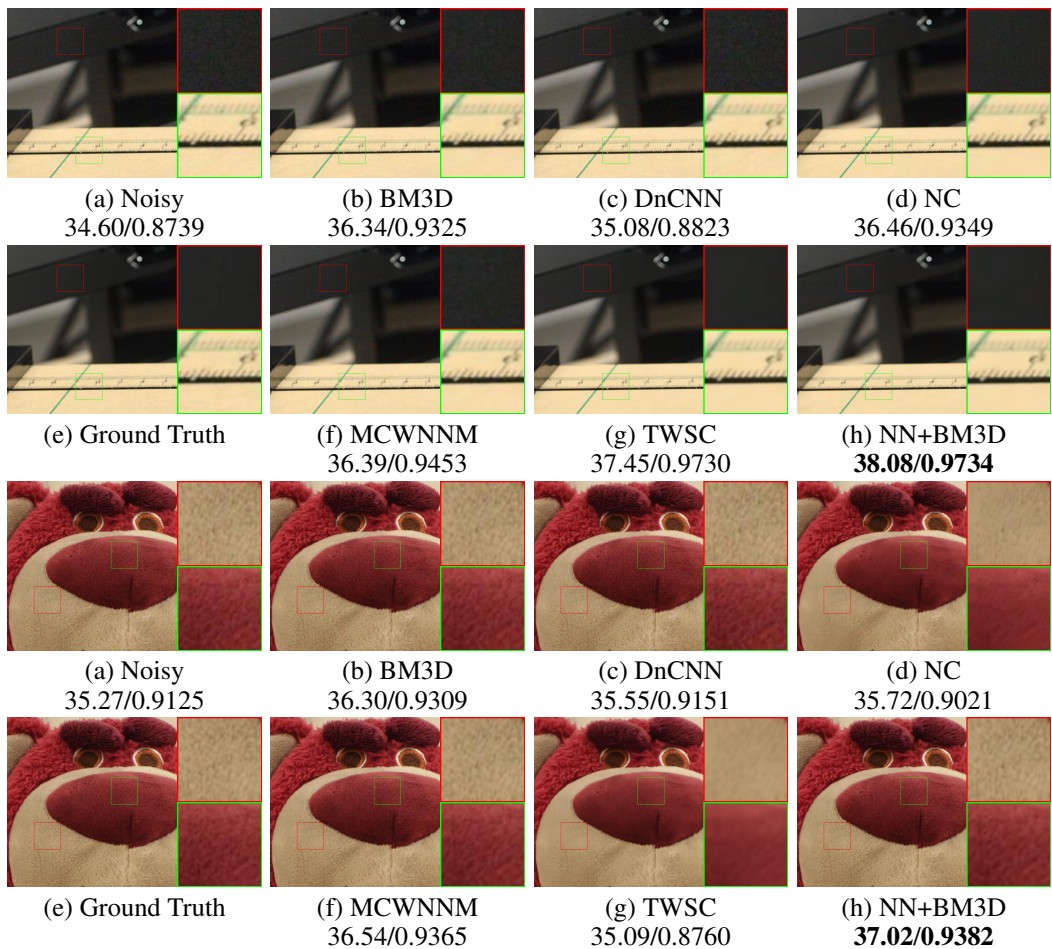

Figure 8: Visual results and PNSR/SSIM of five noisy images from PolyU.

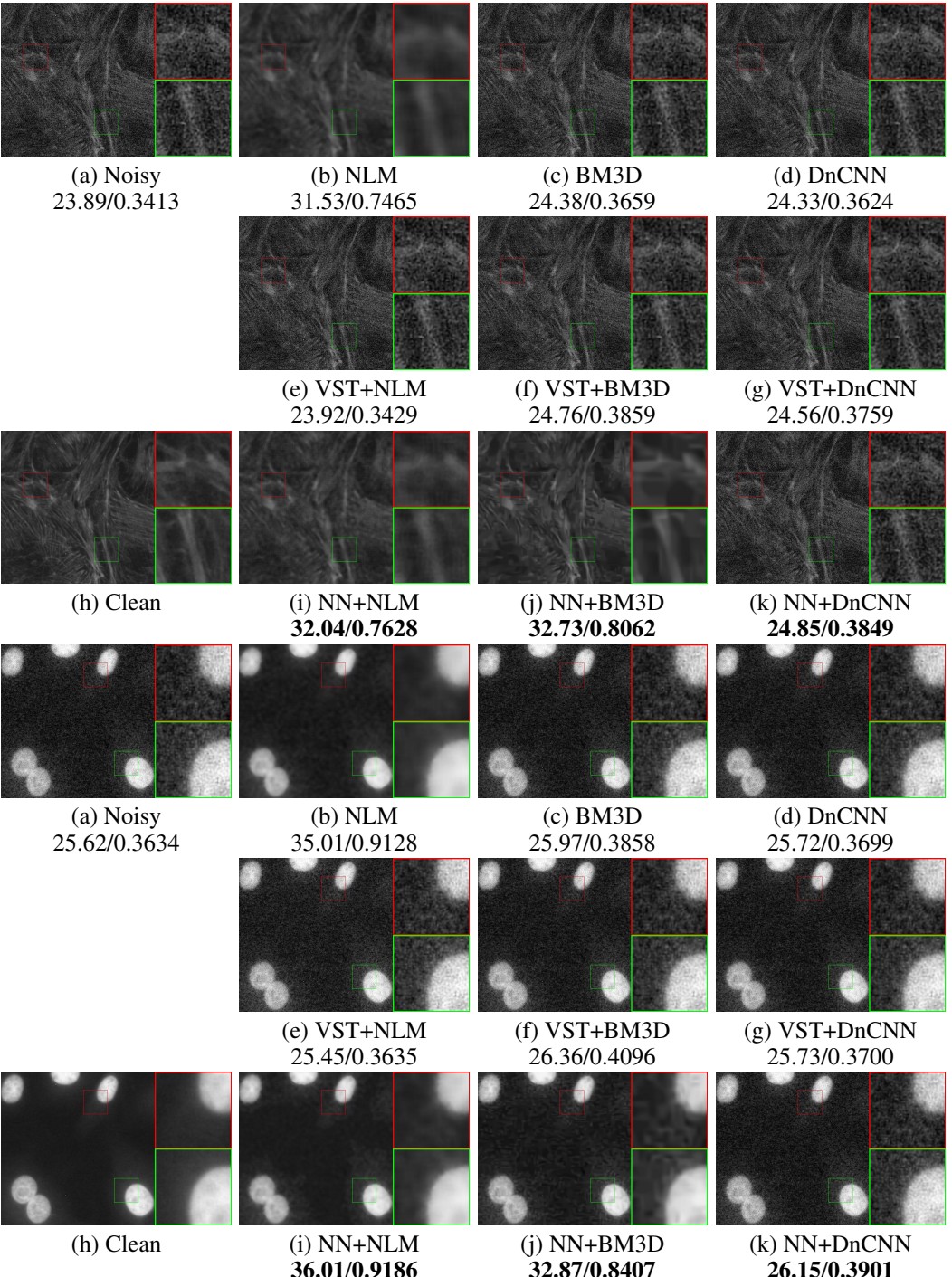

Figure 9: Visual results and PNSR/SSIM of two noisy images from FMDD.

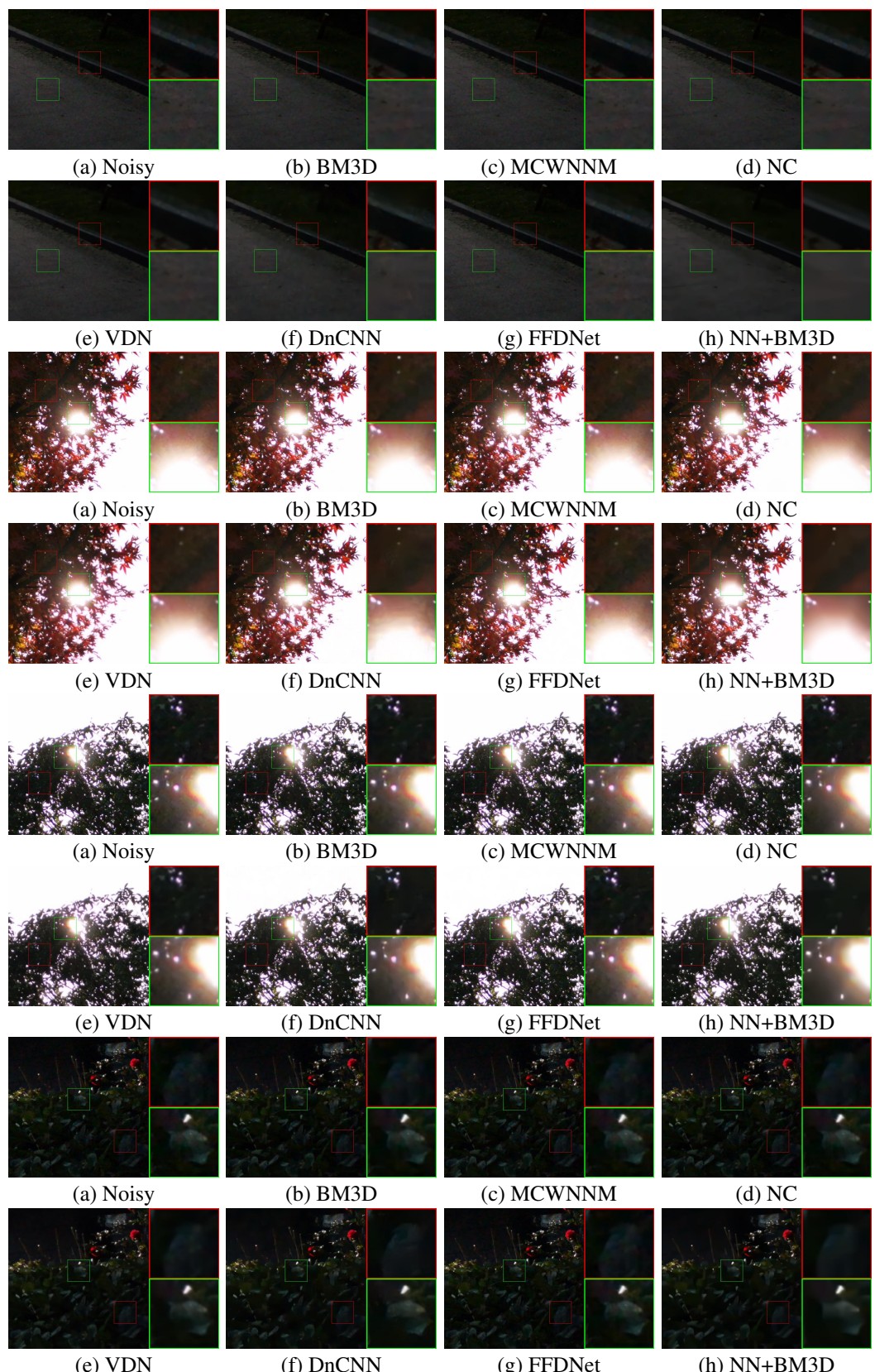

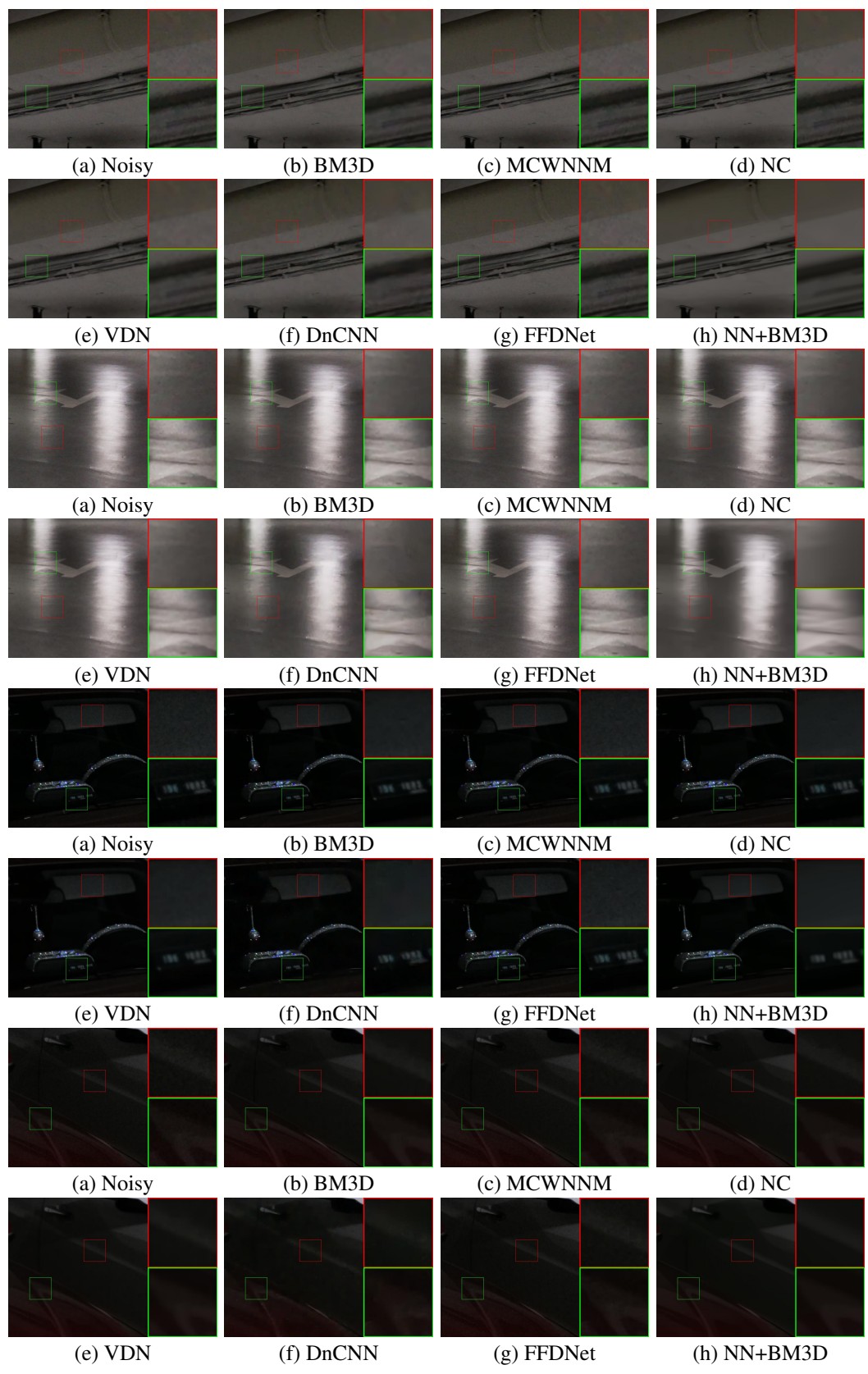

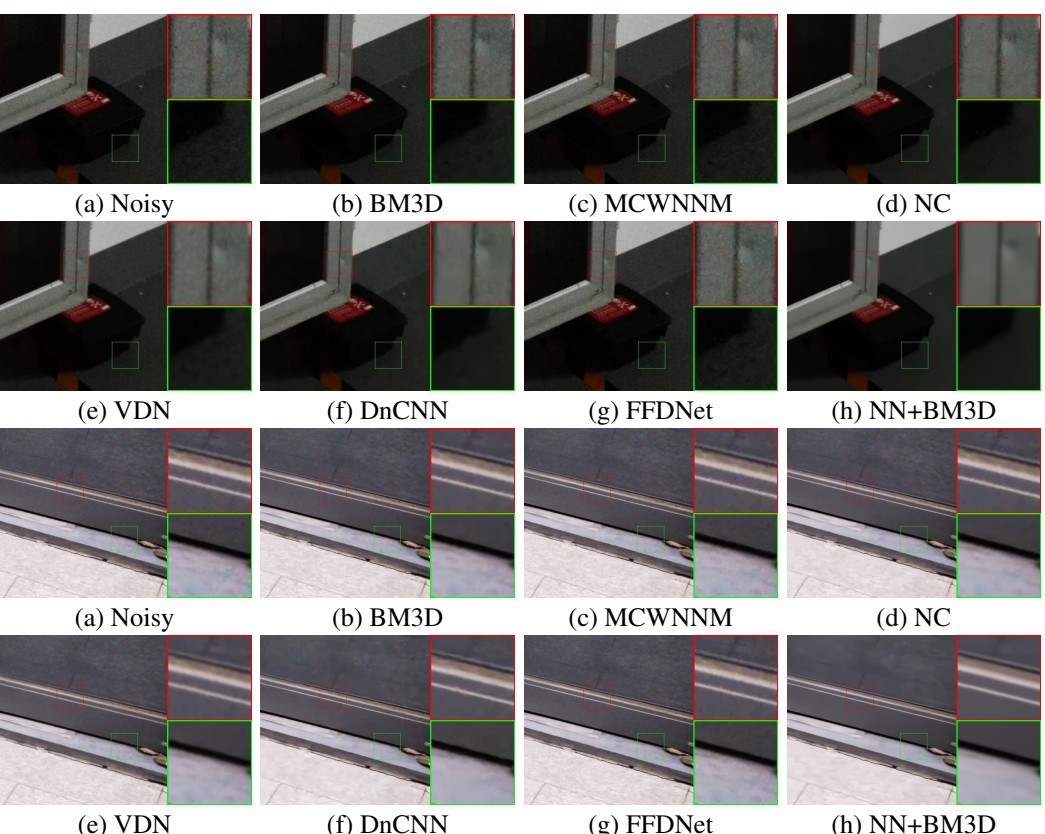

Figure 10: Visual results for real-world image denoising.

