# OpenReview forum: "An Unsupervised Deep Learning Approach for Real-World Image Denoising"
_ICLR.cc/2021/Conference — ICLR 2021 Poster_

### Official Review · AnonReviewer3 · 2020-10-27
**An interesting paper but some points need clarification**

**Rating:** 7
**Confidence:** 4

**Review:**

The key idea of this paper is to transform a real-world noisy image into a latent image space using an encoder neural network(NN), where the noise is hopefully white Gaussian in the latent image space so that existing Gaussian denoisers can be used in the optimization steps. The proposed methods are reported to have improved performance compared to original Gaussian denoisers combined in their methods. One specific proposed combination, i.e., NN+BM3D shows better results than other unsurprised denoising approaches and is competitive with a supervised network. The idea of this paper is interesting and the presentation of the major idea is relatively clear.

However, the reviewer suggests that the authors address the following questions and issues to make the paper clearer:
1. Near equation (1), please define $y$, i.e., an input noisy image.
2. In (2), I think a $\lambda$ is missing and the equal sign does not hold. There should be a constant. Further, seem that the dimension of an image, i.e., $N$, is also used as an iteration number in Algorithm 1. Please avoid this confusion, unless there is a reason to do so.
3. Before Remark 1, the authors mention that the sampling number is equal to 1. I think it would be better to mention when this sample is taken in Algorithm 1. Currently, in my guess, I think $\epsilon$ is re-sampled every time before backpropagation is made so that it looks like a "stochastic" step, however, there are also other possibilities. Please clarify this.
4. For Algorithm 1, there is a significant concern for initializing network parameters $\theta_1$ and $\theta_2$. Are they randomly initialized for your optimization algorithm, or they are inherited from some pre-trained models? I have this concern because in my mind, when updating $x$, it should not go too far away from $y$, and it gradually converges to the final results. If $\theta_1$ and $\theta_2$ are randomly initialized, $x$ might be very different from $y$ in the first several steps and the convergence is not very straightforward. However, if it can still converge, then the robustness of this algorithm can be demonstrated. Please clarify how these parameters are initialized.
5. I recommend that the authors show the latent image and denoised image in $k_{\rm th}$ iteration step for different $k$ in Algorithm 1. The results for NN+BM3D and one noisy image could be enough. For example, the authors can show the latent image at the first several iteration steps and the latent image when the algorithm tends to converge.
6. Are the results of DnCNN+, FFDNet+, CBDNet, etc., directly copied from other papers, for example, Hou et al. (2019)? If so, please highlight this, for example, with footnotes. Further, please clarify why the results for the proposed method in Hou et al. (2019) are not compared.
7. It is better to mention the running time in section 4 or 5, rather than in the Appendix.

-----
After rebuttal:

I think the authors well-addressed my comments. Thanks for their efforts. I would suggest that the authors include some results in their response in the paper or supplementary results if they have not done it.

---

### Official Review · AnonReviewer2 · 2020-10-28
**Interesting idea, good results, not enough information about practical compute concerns**

**Rating:** 8
**Confidence:** 5

**Review:**

##Updated Review##

I'd like to thank the authors for their comments, clarifications and modifications.  I still believe that this paper is a novel contribution with clear impressive results.

# The main idea:
Developing an unsupervised imagedenoising approach is difficult because the real world noise distribution is much more complex than the simplified AWGN assumptions. The authors address this by mapping the noisy image into a latent space in which. AWGN assumption holds, and thus any existing gaussian denoiser is applicable.  The authors claim that the network can be trained with the input image itself and does not require any pertaining in other datasets.

This is a clever idea, it essentially is MAP gaussian denoising with a variational auto encoder as the likelihood model.

Since it is an image-specific network it reduces the need for wide-space sampling fo the data distribution.  How fast is it?  The authors optimize with ADMM iteratively updating network weights and image estimates (which can be time consuming).  No mention is made of compute time when compared to comparable methods.

The encoder and decoder network shapes and architectures are chosen without much reasoning given other than that they are standard choices.  Did the authors explore any hyper parameter sweeps (of the Unet architectures specifically) in this space to see if performance was further improved or impoverished by different choices?

Strength:
Can be utilized with other existing gaussian denoisers to improve their performance.

Visual Results are high quality and show clear improvement for baseline methods.

Weaknesses:
The visual improvement for NN+DnCNN is debatable though the other results show significant improvement. (At least in the chosen examples)

The following is stated as fact but is not backed up with data.

It is noted that there is an expectation term in (4) that can be estimated by the Monte Carlo method and there is no need for a large amount of samples in practice. In our experiments, the sampling number is equal to 1.


Please fix:
On page 1 you write 'Due to the violence of the AWGN assumption” I assume this is supposed to read “Due to the violation”.

Also, on page 5 you write :
"Remark 1 In our model, the basic assumption is that the latent image z is a white Gaussian per- turbations of clean image x, i.e. z|x ∼ N (x, σ2I). Since z = Gθ2 (y), the second term in (4) can be seen as the transformed data fidelity while the first term is a regularization such that the encoder avoids from the trivial mappings, e.g. identity map or zero map."

The first term, which is  1Eε||Fθ1(Gθ2(y)+ε)−y||2  does prevent a zero mapping, but does not on it's own prevent the identity map (it is in fact encouraging it).  However, the combination of the first, second ( 1 ||Gθ2(y)−x||2) and third (λR(x)) is what pushes against the degenerate identity mapping solution.  Please update this sentence to be more correct.

---

### Official Review · AnonReviewer4 · 2020-10-30
**Review of An Unsupervised Deep Learning Approach for Real-World Image Denoising**

**Rating:** 6
**Confidence:** 4

**Review:**

This paper proposes a method to perform denoising using a single image. A UNet encoder is used to map the image to a space where a denoiser such as BM3D is applied. ADMM is used to update the weights by minimizing a cost function where one of the terms fits the noise image (using an additional UNet decoder trained jointly) and the other term is associated to the denoiser. The method yields promising results.

*Quality*:

The methodology proposed by the paper is intriguing, but I don't think it is analyzed adequately. The Bayesian motivation for the method is too detached from the algorithm that is finally applied. For example, the authors explain the method using expectations and then at some point state the following:

"It is noted that there is an expectation term in (4) that can be estimated by the Monte Carlo method and there is no need for a large amount of samples in practice. In our experiments, the sampling number is equal to 1."

This makes no sense. In fact, in Algorithm 1 we see that the $\epsilon$ term from (4) does not appear. Also, the authors state

"the proposed NN approximates the likelihood in the classic Bayesian framework, which gives clear interpretations of each loss term"

but show no evidence that the terms are actually meaningful. Due to the lack of analysis, the method seems very ad hoc. For example, it is unclear what role the decoder has. How do the results change with different choices of encoder/decoder? It would be very helpful to add some ablation studies.

Also, the authors show that the method is stable with respect to the hyperparameters, but it seems to be that there should be a meaningful dependence with $\sigma$. How should it be estimated if you want to train on a single image, where the noise is not Gaussian? In the experimental section, they state: "The noise level is estimated from Donoho & Johnstone (1994); Chen et al. (2015a)". Is this related to $\sigma$? I don't understand what the authors mean.

*Clarity*:

The paper is not written clearly. It took me a while to understand what the authors were actually doing.

*Significance*:

It seems to me that the authors have designed a promising method, but unfortunately they do not provide enough insight and analysis to show why it may work, or what the limitations may be. This will limit the possible impact and usefulness of the paper significantly.

*Pros*:

The method is interesting and novel to the best of my knowledge. It produces promising results.

*Cons*:

Not much analysis. Confusing motivation. Lack of clarity.

Updated review: I appreciate the authors' response and have updated my rating. However, I still believe that the clarity of the exposition could be improved.

---

### Official Review · AnonReviewer1 · 2020-10-30

**Rating:** 6
**Confidence:** 3

**Review:**

The paper proposes an unsupervised approach for denoising for which works on a single noisy image. The central idea is to use a neural network to map the noisy image to a latent image space where the noisy distribution follows Gaussian (like Variance Stabliziation Transform) and then use a off-the-shelf Gaussian denoiser on this image. The paper shows good numerical results.

1. The details of the algorithm is not clear to me. Suppose your noisy image, y has pure Poisson noise. When you use Algorithm 1 and try to apply Gaussian denoiser to $x^{k+1}+q^{k}/\rho$ does it still work? In my experience, these algorithms, particularly DnCNN is very sensitive to the type of the noise and I wonder how much of this algorithm holds when the noise distribution is significantly different from Gaussian. The test cases (including the synthetic noise) here seems to be very close to Gaussian. Authors should produce evidence that the algorithm can actually work on noise distribution significantly different from Gaussian distribution.

2. The paper does not compare to some relevant baselines including:
a) "Self2Self With Dropout: Learning Self-Supervised Denoising From Single Image" Yuhui Quan, Mingqin Chen, Tongyao Pang, Hui Ji; Proceedings of the IEEE/CVF Conference on Computer Vision and Pattern Recognition (CVPR), 2020, pp. 1890-1898
This paper introduces a denoising method which works on a single noisy image and is agnostic to the noise distribution.
b) "High-Quality Self-Supervised Deep Image Denoising" Samuli Laine, Tero Karras, Jaakko Lehtinen, Timo Aila, NeurIPS 2019
has better empirical results that the N2N baseline used in the paper


Update after rebuttal: I appreciate the authors response and have updated my score. Please see some lingering thoughts below:

1. It looks like the experiments show that just the NN based Gaussianization of latent space is not effective when the noise distribution is significantly different from Gaussian. The authors had to  introduce an additional VST in the latent space to make it work. I think this is a limitation of the work - the method just by itself is not capable of handle of noise type significantly different from Gaussian. In my opinion this should be acknowledged in the paper.

2.  I think it is unfair to use a pretrained HQS model for comparison. The authors should retrain the model on the particular datasets they are interested in.

---

### Decision · Program_Chairs · 2021-01-07
**Final Decision**

**Decision:**

Accept (Poster)

**Comment:**

After reading the author’s response, all reviewers recommend accepting the paper. R2 and R3 strongly support the paper while R1 and R4 consider it borderline.

There is agreement that the idea of the work is interesting and novel. The experimental results look solid.

The authors provided an extensive response addressing most of the concerns of the reviewers. In light of this feedback, the reviewers provided some additional comments (which the authors could not address, as the discussion period was over). The AC considers that the authors should incorporate this feedback to the final version of the manuscript. Specifically,

Responding to R1's first question regarding the noise distribution on the original image being significantly different from Gaussian. The authors provided detailed results, which is highly appreciated. As R1 points out, the authors had to introduce an additional VST for the method. These results should be added to the manuscript is important, to show the limitations of the approach.

R3 asks about the importance of the initialization of the weights of the encode and decoder. This is a natural question as this is a non-convex problem. The authors clarify in the manuscript the initialization of x, but do not comment on the weights. It would be good to add a sentence in this regard (as done in the discussion).

R4 mentioned, and the AC agrees, that the authors should try to improve the clarity of the exposition.

The AC considers it important to add in the appendix more visual examples to quantitatively show the performance of the method.